# NaturalSigner: Diffusion Models are Natural Sign Language Generator

## Abstract

Generating natural and expressive sign language pose sequences from text has important practical significance. However, current sign language generation (SLG) methods suffer from low quality and limited expressiveness. In this work, we propose **NaturalSigner**, a classifier-free diffusion-based generative model designed specifically for SLG. Specifically, it consists of a mixed semantic encoder that enhances the semantic consistency and expressiveness of the generated sign language, which takes both text and gloss as input; and a novel sign language denoiser that generates natural sign language pose sequences according to the output of the semantic encoder. In addition, to achieve more natural and high-quality SLG, we design a sign language prompting mechanism to facilitate in-context learning in the diffusion model and duration predictor. Experiments on two datasets show that NaturalSigner significantly outperforms the state-of-the-art methods in terms of semantic consistency, naturalness, and expressiveness. On the Phoenix-2014T dataset, compared with the previous best end-to-end SLG method, NaturalSigner improves the BLEU-4 score of the back translation metric by more than **40%** and reduces the Frechet Inception Distance (FID) by more than **12 times**. Our code and evaluation models are provided in the anonymous link[1].

## 1 Introduction

Sign language is a rich visual language with complex grammatical structures, and it is the main means of communication for nearly 466 million hearing-impaired people worldwide (Organization., 2023). Sign language generation (SLG) technology has great social significance. For example, it can help hearing-impaired people better integrate into society and communicate with others, and facilitate them to receive the latest information (e.g., sign language broadcasts of news and weather forecasts). Due to various reasons, SLG is a challenging task, including the diversity of possible sign language motions, the scarcity of high-quality sign language data, etc. A word may have multiple different sign language motions, and the same sign language motion may also be performed differently by different signer. For example, in Chinese Sign Language, there are two different sign language motions for "patriotism," and the sign language motions for "love" also differ when expressing the meanings of "romantic love" and "hobby/interest." All these constitute a complex one-to-many mapping problem.

Current methods have made some progress in this field, demonstrating plausible mapping from text to sign language pose sequences (Huang et al., 2021; Saunders et al., 2020b; 2022; Stoll et al., 2020; Saunders et al., 2020a; 2021a;b). However, the generated results of these methods still suffer from low quality, lack of naturalness, and limited expressiveness. Since SLG is a one-to-many mapping problem, some previous methods that use regression-based models to learn this mapping often results in over-smoothing and blurry outputs (Saunders et al., 2020a), which we call the "mean sign pose" problem. In addition, most of the previous SLG methods adopt autoregressive models, and due to the error accumulation problem, the semantic consistency and expressiveness of the generated sign language pose sequences are insufficient. (Saunders et al., 2020a) attempted to alleviate the "mean sign pose" problem by introducing adversarial training. However, we believe that diffusion models are better candidates for the SLG problem, as they can well express the one-to-many mapping problem we described, and have advantages in terms of training difficulty and generation performance.

---

[1] https://anonymous.4open.science/r/NaturalSigner

In this paper, we propose naturalSigner, a carefully adapted diffusion based generative model for SLG problem, which can achieve semantically consistent, natural, and highly expressive sign language generation. As shown in Figure 1, to generate more faithful results, we first carefully design a mixed semantic encoder, which can fuse information from both gloss and text to generate higher quality prior vectors. Secondly, we propose a novel diffusion model, which generates sign language pose sequences according to the prior vectors. Specifically, to maintain the semantic consistency between the generated results and the prior vectors in the time dimension, we introduce the condition by directly adding the prior vectors to the hidden states, rather than using cross-attention. As sign language motions are closely related to the local neighboring frames, we replace the feedforward neural network (FFN) layer of the Transformer with one-dimensional convolution. Finally, to encourage the diffusion model to follow the characteristics in the sign language prompt, we design a sign language prompting mechanism to facilitate in-context learning in the diffusion model and duration predictor.

Benefiting from these designs, NaturalSigner has significant improvements over the state-of-the-art methods in terms of semantic consistency, naturalness, and expressiveness. We evaluate our model on two commonly used SLG datasets, Phoenix-2014T (Camgoz et al., 2018) and Phoenix-2014 (Koller et al., 2015). Regarding semantic consistency, experiments on the Phoenix-2014T dataset show that our method achieves more than **15%** and **40%** relative improvements over the state-of-the-art end-to-end SLG method in terms of back translation metrics ROUGE-L (Lin & Och, 2004) and BLEU-4 (Papineni et al., 2002) scores. In terms of naturalness and expressiveness, our method reduces the Frechet Inception Distance (FID) (Heusel et al., 2017) by more than **12 times** compared with the state-of-the-art method, and the diversity (Lee et al., 2019) of the generated results is also better. Samples of sign language pose sequences synthesis can be found in the web demo page[2].

## 2 RELATED WORK

### 2.1 SIGN LANGUAGE GENERATION

Sign language Generation (SLG) models take input text or gloss and convert them into pose sequences (e.g., Progressive Transformers(Saunders et al., 2020b), MOMP(Saunders et al., 2021b)), which are then translated into videos using a video renderer. Directly generating sign language videos from text has not been successfully practiced. SLG has gradually developed from generating isolated signs (Stoll et al., 2018; Zelinka et al., 2019) and then connecting them into sequences to end-to-end continuous SLG, which directly regressed sequences of multiple signs. Progressive Transformers (Saunders et al., 2020b) is the first end-to-end continuous SLG model, which uses a transformer-based encoder-decoder structure, where the decoder generates sign language pose sequences in an autoregressive manner. This work introduces a counter decoding technique to predict the length of pose sequences by tracking the production progress, hence the name Progressive Transformers. Adversarial Multi-Channel SLG (Saunders et al., 2020a) introduces an adversarial discriminator to improve the quality of generated sign language poses. Mixture Density (Saunders et al., 2021a) combines Transformers and mixture density networks to model multimodal continuous sequences. MOMP (Saunders et al., 2021b) proposes a transformer-based Mixture-of-Experts architecture, which combines learned motion primitives at the frame level to enhance the generation performance. FS-NET (Saunders et al., 2022) alleviates the error accumulation problem of the above autoregressive SLG models(Saunders et al., 2020b;a; 2021b;a) and the "mean sign pose" problem of regression-based models (Saunders et al., 2020b; 2021b) by learning the optimal temporal alignment between dictionary and continuous sign sequences. However, it requires additional high-quality isolated sign language dictionaries, and the naturalness and expressiveness of the generated sign language pose sequences are limited. In this work, we propose NaturalSigner, a diffusion generative model specifically designed to tackle the challenges of error accumulation and mean sign pose.

### 2.2 DIFFUSION GENERATIVE MODELS

A diffusion probabilistic model is a parameterized Markov chain trained by optimizing variational lower bound, which generates samples matching the data distribution in constant steps (Ho et al., 2020). Diffusion model is first proposed in (Sohl-Dickstein et al., 2015). Then DDPM (Ho et al.,

---

[2] https://naturalsigner.github.io/

2020) makes progress in high-quality image generation and reveals an equivalence between diffusion model and denoising score matching (Song & Ermon, 2019; Song et al., 2020). After success in unconditional generation, these models are extended to work in conditional generation settings, demonstrating competitive or even better performance than GANs. For conditioned generation, (Dhariwal & Nichol, 2021) introduced classifier-guided diffusion, which was later on adapted by GLIDE (Nichol et al., 2021) to enable conditioning over CLIP (Radford et al., 2021) textual representations. Subsequently, (Ho & Salimans, 2022) proposed a ClassifierFree Guidance approach that enables conditioning without requiring pretraining of the classifiers. In addition to the great success in image generation, diffusion models have also been successfully applied in other fields, such as speech synthesis (Kong et al., 2020), video generation (Ho et al., 2022), singing voice synthesis (Liu et al., 2022), etc. In this paper, we implement text-conditioned sign language generation in a classifier-free manner.

## 3 NATURALSIGNER

In this section, we introduce NaturalSigner, a diffusion-based generative model that can generate natural and expressive sign language pose sequences from text. We introduce the sign language representation in subsection 3.1, and then the sign language prompting mechanism in subsection 3.2. Finally, we introduce the details of our diffusion model and duration predictor in subsection 3.3 and subsection 3.4.

### 3.1 SIGN LANGUAGE REPRESENTATION

To better represent the complex body movements in sign language, we propose to use the pose parameter $\vec{\theta} = \left[\vec{\omega}_0^T, \ldots, \vec{\omega}_K^T\right]^T$ of the SMPL-X human body model (Pavlakos et al., 2019) as the sign language representation, instead of the 3D joint coordinates in Euclidean space used in previous works. Where $\vec{\omega}_k \in \mathbb{R}^3$ denotes the axis-angle representation of the relative rotation of part $k$ with respect to its parent in the kinematic tree. However, since the axis-angle form is not a continuous rotation representation, which is not conducive to network learning, we further convert it to the rotation 6D representation (Zhou et al., 2019) $\vec{o} = \left[\vec{r}_0^T, \ldots, \vec{r}_K^T\right]^T$, $|\vec{o}| = 43 \times 6 = 258$. We ignore the lower body joints outside the visible range, and the detailed information of the joint selection and sign language representation can be found in Appendix A. There are three advantages of using this representation: 1) it has rotation and translation invariance; 2) it separates the modeling of human body shape and pose, and the semantics of sign language should only be related to the pose and independent of the shape; 3) the introduction of human body prior avoids generating abnormal results, such as fingers longer than arms.

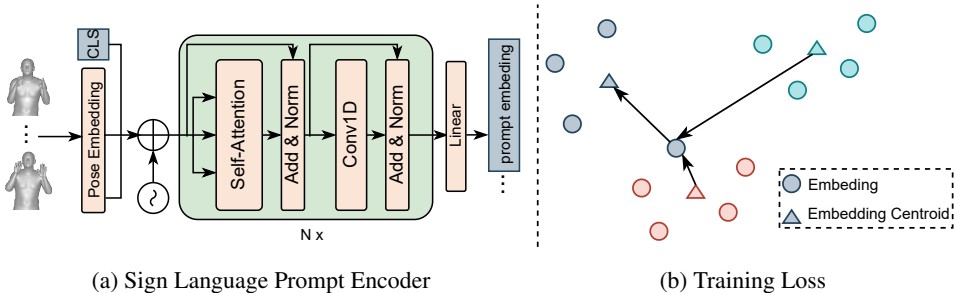

| (a) Sign Language Prompt Encoder | (b) Training Loss |

Figure 1: The structure and training loss of sign language prompt encoder. In Figure 1a "sinusoidal-like symbol" denotes the positional encoding. In Figure 1b, embeddings with the same color represent prompt embeddings from the same signer. During training, we pushes each embedding vector close to its centroid and pull it away from all other centroids.

### 3.2 SIGN LANGUAGE PROMPTING FOR IN-CONTEXT LEARNING

To facilitate in-context learning for generating more natural sign language pose sequences, we design a sign language prompting mechanism to encourage the diffusion model and duration predictor to

follow various information in the sign language prompt (e.g., signer identity). For any sign language pose sequence, we randomly sample a sequence from the remaining sign language pose sequences of the corresponding signer as the sign language prompt. As shown in Figure 1, we use a Transformer-based (Vaswani et al., 2017) sign language prompt encoder to encode the sign language prompt, and use a learnable CSL token to summarize the encoded features to obtain the prompt embedding. In particular, we replace the 2-layer dense network in the vanilla Transformer with a 2-layer 1D convolutional network (Ping et al., 2018; Ren et al., 2019; Gehring et al., 2017), because the adjacent inputs in sign language are more closely related(Aoxiong et al., 2023). As shown in Figure 2, to utilize this prompt embedding as a prompt, for the duration predictor, we concatenate it to the input text embedding, and for the diffusion model, we directly add it as a token to the input. Since the amount of sign language data is limited, inspired by (Wan et al., 2018), to encourage the learning of useful prompt information, we add an auxiliary prompt loss.

$$\mathcal{L}_{prompt} = \sum_{i=1}^{N} -log\frac{\exp(w \cdot cos(e_i, c^i) + b)}{\sum_{k=1}^{M} \exp(w \cdot cos(e_i, c_k) + b)} \tag{1}$$

where $e_i$ is the prompt embedding of the $i$-th sign language pose sequence, $c^i$ represents the centroid of the embedding of the $i$-th sequence, $c_k$ is the centroid of the $k$-th signer, $M$ is the number of signers, and $N$ is the number of sign language pose sequence involved in the training. An intuitive explanation of the prompt loss is shown in Figure 1b. During the training process, we train the encoder to minimize the distance between each embedding vector and its corresponding signer centroid while simultaneously maximizing the distance between each embedding vector and all other signer centroids.

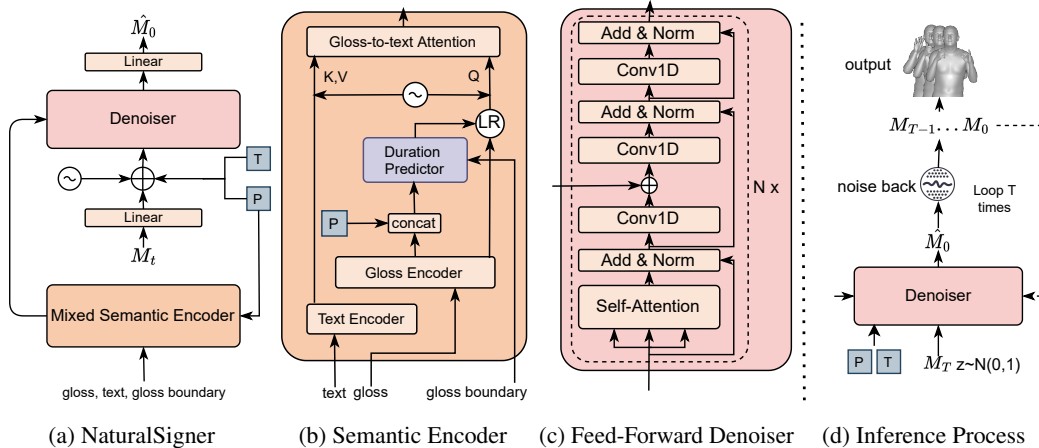

| (a) NaturalSigner | (b) Semantic Encoder | (c) Feed-Forward Denoiser | (d) Inference Process |

Figure 2: The overall architecture for NaturalSigner. In Figure 2a, "P" represents sign language prompt embedding, "T" represents timestep embedding, $M_t$ represents sign language motion at t-th step in the diffusion process, "sinusoidal-like symbol" denotes the positional encoding. In Figure 2b, "LR" denotes length regulator. In Figure 2d, $M_T$ represents motion at T-th diffusion step (Gaussian white noise), at each step denoiser predicts the clean sample $\hat{M}_0$, then noise it back to $M_{t-1}$, loop T times, and finally output $M_0$.

### 3.3 PRIOR MODEL: MIXED SEMANTIC ENCODER AND DURATION PREDICTOR

To obtain rich semantic information and thus generate faithful and high-quality sign language pose sequence, we design a mixed semantic encoder, as shown in Figure 2b, which consists of a spoken language text encoder, a gloss encoder, a duration predictor, and a gloss-to-text attention module. First, we use two encoders to encode spoken language text and gloss into hidden states $H_t$ and $H_g$, respectively. Then we concatenate the prompt embedding $P$ and $H_g$ and input them into the duration predictor to predict the duration of each gloss.

As shown in Figure 3, the duration predictor consists of a 2-layer 1D convolutional network with ReLU activation, each followed by the layer normalization and the dropout layer, and an extra linear layer to output a scalar, which is exactly the predicted gloss duration. Then, we use a length regulator to duplicate and expand sequence $H_g$ into sequence $H_{exp\_g}$ based on the predicted duration sequence $D$, in order to address the issue of mismatch between the length of gloss and sign language motion. For example, if $H_g = [h_1, h_2, h_3]$ and $D = [2, 1, 3]$, then $H_{exp\_g} = [h_1, h_1, h_2, h_3, h_3, h_3]$. Finally, to add more semantic information, we introduce a gloss-to-text attention module, which takes $H_{exp\_g}$ as the query, $H_t$ as the key and value, and outputs a semantically fused hidden state $H_{g\_fused}$. The training process of the duration predictor is shown in Figure 3.

We use the state-of-the-art continuous sign language recognition (CSLR) model (Chen et al., 2022) to extract the alignment between gloss and sign language video as the ground-truth duration. First, we use the CSLR model to extract the connectionist temporal classification (CTC) (Graves et al., 2006) probability matrix of the gloss and sign language video, and then we use the Viterbi algorithm to search for the optimal alignment(Zhou et al., 2021). Finally, we use the mean square error between the ground-truth duration and the predicted duration as the loss for training.

$$\mathcal{L}_{duration} = ||D - \hat{D}||_2^2 \tag{2}$$

where $D$ and $\hat{D}$ represent the ground-truth duration and the predicted duration respectively.

Figure 3: The structure and training process of the duration predictor.

## 3.4 FEED-FORWARD DENOISER

The architecture for the Denoiser is a feed-forward structure based on self-attention in Transformer (Vaswani et al., 2017) and 1D convolution. We call this structure as Feed-Forward Denoiser (FFD), as shown in Figure 2c. Each FFD block contains a self-attention network, a semantic fusion network, and a 1D convolution network. The self-attention network is used to learn long-distance dependency relationships in the sequence. Different from vanilla Transformer using cross-attention to fuse semantic information, our semantic fusion network consists of two 1D convolution networks and directly uses the addition operation instead of the attention operation. The motivation is that the hidden state $H_{g\_fused}$ output by the semantic encoder has been aligned with the sign language motion, so there is no need to use a complex attention mechanism for alignment and fusion. Using our designed fusion method can reduce the computational complexity to linear while avoiding the alignment information being disrupted. Following Transformer (Vaswani et al., 2017), residual connections, layer normalization, and dropout are added after the self-attention network, semantic fusion network, and 1D convolutional network respectively.

As for the training process of the denoiser, we follow the framework in denoising diffusion probabilistic models (Ho et al., 2020). Suppose $M_0$ is a real data sampled from the sign language motion data distribution, diffusion is modeled as a Markov noising process $\{M_t\}_{t=0}^T$, where $M_t$ is a noisy version of $M_0$, and:

$$q(M_t|M_{t-1}) = \mathcal{N}(\sqrt{\alpha_t}M_{t-1}, \sqrt{1-\alpha_t}I) \tag{3}$$

where $\alpha_t \in (0, 1)$ is a constant hyperparameter. When $\alpha_t$ is small enough, we can approximate $M_T \sim \mathcal{N}(0, I)$. The distribution $p(M_0|c)$ that needs to be modeled for conditional sign language motion synthesis can be obtained by the reversed diffusion process from $M_T$ to $M_0$. Finally, we use the following loss for training:

$$\mathcal{L}_{\text{diff}} = \mathbb{E}_{M_0 \sim q(M_0|c), t \sim [1,T]} \left[ ||M_0 - D(M_t, t, c)||_2^2 \right] \tag{4}$$

where $D$ is the denoiser network, $c$ is the condition for generating sign language motion, including the output of the semantic encoder and the prompt embedding. Similar to (Ramesh et al., 2022; Li et al., 2022), in order to achieve better results, we let the denoiser network predict $M_0$. The total loss function for the diffusion model is as follows:

$$\mathcal{L} = \lambda_1 \mathcal{L}_{\text{diff}} + \lambda_2 \mathcal{L}_{\text{prompt}} + \lambda_3 \mathcal{L}_{\text{duration}} \tag{5}$$

Table 1: The experimental results of back translation metric on Phoenix-2014T dataset. For convenience, we use R to represent ROUGE-L, B1→B4 to represent BLEU-1→BLEU-4.

(a) Back translation results for the *Gloss* to *Pose task*.

| Method | Extra Data | DEV | | | | | TEST | | | | |
|---|---|---|---|---|---|---|---|---|---|---|---|
| | | R↑ | B1↑ | B2↑ | B3↑ | B4↑ | R↑ | B1↑ | B2↑ | B3↑ | B4↑ |
| PT Saunders et al. (2020b) | n/a | 34.01 | 32.40 | 20.50 | 15.08 | 11.93 | 32.02 | 31.80 | 19.19 | 13.51 | 10.43 |
| Multi-Channel SLG Saunders et al. (2020a) | n/a | 36.75 | 34.09 | 22.42 | 15.52 | 13.16 | 34.19 | 32.41 | 20.95 | 15.31 | 12.16 |
| Mixture Density Saunders et al. (2021a) | n/a | 39.06 | 33.84 | 22.59 | 16.77 | 13.14 | 35.19 | 33.66 | 21.19 | 15.22 | 11.94 |
| MOMP Saunders et al. (2021b) | n/a | 37.58 | 34.21 | 22.67 | 16.71 | 13.32 | 35.61 | 33.95 | 22.02 | 16.03 | 12.67 |
| **NaturalSigner (ours)** | n/a | **42.51** | **41.07** | **29.65** | **23.44** | **19.55** | **41.97** | **41.13** | **29.86** | **23.52** | **19.54** |
| Dictionary Sequence Saunders et al. (2022) | dictionary | 38.11 | - | - | - | 16.28 | 36.95 | - | - | - | 16.27 |
| FS-NET Saunders et al. (2022) | dictionary | 40.94 | - | - | - | 19.14 | 40.60 | - | - | - | 18.78 |

(b) Back translation results for the *Text* to *Pose task*.

| Method | Extra Data | DEV | | | | | TEST | | | | |
|---|---|---|---|---|---|---|---|---|---|---|---|
| | | R↑ | B1↑ | B2↑ | B3↑ | B4↑ | R↑ | B1↑ | B2↑ | B3↑ | B4↑ |
| PT Saunders et al. (2020b) | n/a | 34.05 | 33.12 | 20.71 | 14.71 | 11.43 | 31.07 | 29.74 | 17.62 | 12.53 | 9.68 |
| Multi-Channel SLG Saunders et al. (2020a) | n/a | 33.68 | 31.84 | 20.58 | 15.61 | 12.65 | 32.74 | 30.93 | 18.99 | 13.72 | 10.81 |
| Mixture Density Saunders et al. (2021a) | n/a | 33.40 | 30.94 | 19.63 | 14.48 | 11.54 | 33.19 | 31.56 | 19.70 | 14.55 | 11.68 |
| MOMP Saunders et al. (2021b) | n/a | 37.76 | 35.23 | 23.49 | 17.50 | 14.03 | 36.77 | 35.89 | 23.27 | 16.86 | 13.30 |
| **NaturalSigner (ours)** | n/a | **43.65** | **41.11** | **30.11** | **23.68** | **19.63** | **44.0** | **41.53** | **30.19** | **23.73** | 19.65 |
| FS-NET Saunders et al. (2022) | dictionary | 35.74 | - | - | - | 16.92 | 42.57 | - | - | - | **21.10** |

# 4 EXPERIMENTS

## 4.1 EXPERIMENTAL SETUP

**Datasets** We evaluate NaturalSigner on the Phoenix-2014T(Camgoz et al., 2018), and Phoenix-2014 (Koller et al., 2015) datasets. Phoenix-2014T is the most commonly used sign language generation dataset, with a vocabulary size of 1066 for glosses and 2887 for German text. Phoenix-2014 is a German SLR dataset with a vocabulary size of 1081 for glosses. For all datasets, we follow the official split of training set, validation set, and test set. We use the open source tool frankmocap [3] (Rong et al., 2021; Joo et al., 2021) to extract the human motion representation mentioned in subsection 3.1.

**Implementation Details** We train NaturalSigner on 1 NVIDIA V100 GPU. To test the generation effect from spoken language text to sign language, as in previous work (Saunders et al., 2022), we use a standard Transformer to train a text-to-gloss translation model. The detailed training hyperparameters and training time of the translation model, sign language prompt encoder, and diffusion model are listed in Appendix B.

**Evaluation Metrics** We follow (Saunders et al., 2020b) to use the back translation metric to evaluate the robustness of SLG, which uses a pre-trained SLT model (Camgoz et al., 2020) to translate sign language back to text and then calculates the BLEU (Papineni et al., 2002) and ROUGE-L (Lin & Och, 2004) scores between the generated text and the original text. For quantitatively evaluate, we train a sign language feature extractor and a text feature extractor under the contrastive loss in a similar way to CLIP (Radford et al., 2021) to generate geometrically close feature vectors for matched sign language and spoken language text. Inspired by (Lee et al., 2019), we propose to use the following metrics: Frechet Inception Distance (FID) (Heusel et al., 2017), diversity, and multimodal distance to evaluate the quality, and diversity of the generated sign language. We introduce the training details of the back translation model and feature extractor, and the specific calculation methods of these metrics in Appendix C.

---

[3] https://github.com/facebookresearch/frankmocap

Table 2: Experimental results the *Gloss* to *Pose task* on the Phoenix2014T dataset. → means results are better if the metric is closer to the real distribution and ± indicates the 95% confidence interval.The range of the interval is multiplied by $10^2$ for ease of data display. +P means using sign language prompting mechanism.

(a) Experimental results on validation set.

| Method | FID↓ | Multimodal Dist ↓ | Diversity → |
|---|---|---|---|
| Real | $0.000^{\pm.00}$ | $0.651^{\pm.00}$ | $0.798^{\pm.23}$ |
| PT Saunders et al. (2020b) | $1.963^{\pm.00}$ | $1.418^{\pm.00}$ | $0.065^{\pm.07}$ |
| NaturalSigner | $0.161^{\pm.43}$ | $1.021^{\pm.26}$ | $0.755^{\pm.08}$ |
| +P | $\mathbf{0.107^{\pm.07}}$ | $\mathbf{0.988^{\pm.22}}$ | $\mathbf{0.782^{\pm.08}}$ |

(b) Experimental results on test set.

| Method | FID↓ | Multimodal Dist ↓ | Diversity → |
|---|---|---|---|
| Real | $0.000^{\pm.00}$ | $0.645^{\pm.00}$ | $0.646^{\pm.17}$ |
| PT Saunders et al. (2020b) | $1.929^{\pm.00}$ | $1.406^{\pm.00}$ | $0.053^{\pm.06}$ |
| NaturalSigner | $0.151^{\pm.23}$ | $1.023^{\pm.23}$ | $0.607^{\pm.07}$ |
| +P | $\mathbf{0.101^{\pm.10}}$ | $\mathbf{0.979^{\pm.20}}$ | $\mathbf{0.631^{\pm.07}}$ |

Table 3: Experimental results the *Gloss* to *Pose task* on the Phoenix2014 dataset.

(a) Experimental results on validation set.

| Method | FID↓ | Multimodal Dist ↓ | Diversity → |
|---|---|---|---|
| Real | $0.000^{\pm.00}$ | $0.583^{\pm.00}$ | $0.775^{\pm.23}$ |
| PT Saunders et al. (2020b) | $1.882^{\pm.00}$ | $1.403^{\pm.00}$ | $0.078^{\pm.12}$ |
| NaturalSigner | $0.143^{\pm.08}$ | $0.888^{\pm.12}$ | $0.736^{\pm.07}$ |
| +P | $\mathbf{0.113^{\pm.14}}$ | $\mathbf{0.861^{\pm.12}}$ | $\mathbf{0.745^{\pm.07}}$ |

(b) Experimental results on test set.

| Method | FID↓ | Multimodal Dist ↓ | Diversity → |
|---|---|---|---|
| Real | $0.000^{\pm.00}$ | $0.587^{\pm.00}$ | $0.666^{\pm.19}$ |
| PT Saunders et al. (2020b) | $1.879^{\pm.00}$ | $1.408^{\pm.00}$ | $0.076^{\pm.12}$ |
| NaturalSigner | $0.137^{\pm.14}$ | $0.877^{\pm.12}$ | $0.632^{\pm.07}$ |
| +P | $\mathbf{0.113^{\pm.11}}$ | $\mathbf{0.854^{\pm.14}}$ | $\mathbf{0.639^{\pm.06}}$ |

## 4.2 COMPARISONS WITH THE STATE-OF-THE-ART

### 4.2.1 QUANTITATIVE COMPARISON

In this section, we compare our model with previous work in the following aspects: 1) generation robustness, measured by calculating back translation and multimodal distance metrics; 2) generation quality, measured by calculating FID and additional user study; 3) generation diversity, measured by calculating diversity metrics. For the metrics other than back translation, we only report the results on the gloss-to-pose setting, and we show more experimental results and analysis in Appendix D.

**Comparison methods.** 1) Progressive Transformers (PT) Saunders et al. (2020b) is the first end-to-end continuous SLG model. 2) Multi-Channel SLG Saunders et al. (2020a) proposes to use a multi-channel architecture based on generative adversarial networks (GANs) to generate continuous sign language. 3) Mixture Density Saunders et al. (2021a) models sign language poses by introducing Mixture Density Networks for Progressive Transformers. 4) Mixture of Motion Primitives (MOMP) Saunders et al. (2021b) proposes a transformer-based Mixture-of-Experts architecture. It is the previous state-of-the-art end-to-end continuous SLG model. 5) FS-NET Saunders et al. (2022) constructs continuous sign language poses by splicing sign language poses in the dictionary and selecting suitable frames through a selection network. It requires an additional sign language dictionary, and essentially it is not a generation model because it only selects sign language poses in the dictionary. We provide its results as a reference.

**Generation Robustness.** Consistent with previous work, we evaluate on both the gloss to pose and text to pose tasks. For fair comparison, we only use gloss as the input of the semantic encoder in this part of the experiment, without inputting prompt embedding and spoken language text. The experimental results of the *back translation metric* are shown in Table 1. Regarding the gloss-to-pose task, our method has shown significant improvement compared to the original state-of-the-art end-to-end continuous SLG method MOMPSaunders et al. (2021b), as demonstrated in Table 1a. In addition, our method also outperforms the Dictionary Sequence [4] and FS-NET Saunders et al. (2022) methods which require the use of additional high-quality isolated signs from sign language lexicons. As shown in Table 1b, in the text to pose task, our method also outperforms MOMP. Another metric that reflects the fidelity of the generated sign language is the *multimodal distance*. As shown in Table 1a and Table 1b, our method achieves the best results.

---

[4]Dictionary Sequence represents the method proposed in the FS-NET paper to construct sign language pose sequences by sign language dictionary interpolation.

**Generation Quality.** *FID* measures the quality of the generated sign language by calculating the statistical distance between the features of the real sign language and the generated sign language. We show the performance of our method on the FID metric in Table 2 and Table 3. Our method achieves the best results on the FID metric, which shows that the sign language generated by naturalsigner is closer to the real sign language than other methods. The user study in subsubsection 4.2.2 also proves that the quality of the sign language generated by NaturalSigner is better than previous methods.

**Generation Diversity.** The *diversity* metric is better if the numerical value is closer to the ground truth. In addition, it is worth noting that it only makes sense to compare diversity when the quality of the generated sign language is similar, and sign language that is wrong but diverse is meaningless. As shown in Table 2 and Table 3, our method achieves better generation quality and diversity metrics, which indicates that our method is better than other methods in generating sign language diversity. The diversity metrics of methods such as Progresive Transformers are poor, because these regression-based methods are difficult to handle the one-to-many mapping in the SLG problem.

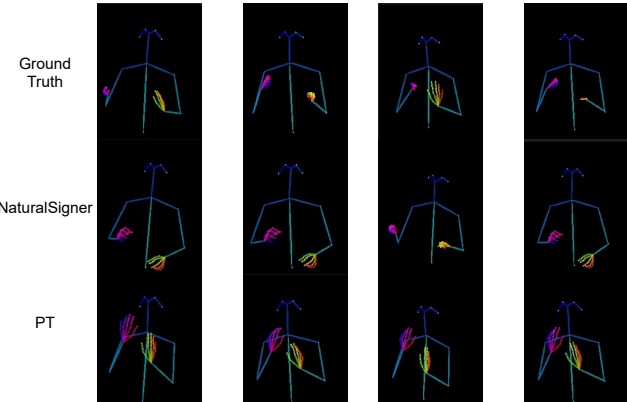

Figure 4: **The comparison of generated key frame results.** Please zoom in for a better visualization and it can be seen that our method is more accurate than the baseline method in both hand details and overall poses. More qualitative results can be viewed on the demo web page.

### 4.2.2 QUALITATIVE COMPARISON

To compare the generation results of the two methods, we present the key frames of the generated sign language gesture sequences in Figure 4. We observe that although there is still room for improvement in both methods, the sign language poses generated by NaturalSigner are closer to the real sign language poses than the baseline method in both hand details and overall poses.

Table 4: Comprehension user evaluation results

| Dataset | NaturalSigner | PT |
|---|---|---|
| Phoenix2014T | **89%** | 11% |
| Phoenix2014 | **90%** | 10% |

**User Study** Similar to the evaluation method in (Saunders et al., 2022), we invited 10 users to participate in the user study and asked them to choose which sign language pose sequence, generated by our method or the baseline approach, was easier to understand and closer to the ground truth. The evaluation results are shown in Table 4, and our method has an overwhelming advantage on both datasets, which indicates that the sign language generated by our method is better than the baseline method in terms of expressiveness and comprehensibility.

### 4.3 ABLATION STUDY AND ANALYSIS

In this section, we introduce the results of our ablation experiments on the Phoenix2014T dataset, and analyze the effectiveness of our proposed method through the experimental results.

**Effectiveness of Sign Language Prompt Mechanism.** As shown in Table 2 and Table 3, on the Phoenix2014T and Phoenix2014 datasets, the FID metrics decreased by more than **33%** and **17%**,

Table 5: Ablation study results on the back translation metric for the *Text* to *Pose task*.

| Method | DEV | | TEST | |
|---|---|---|---|---|
| | ROUGE-L↑ | BLEU-4↑ | ROUGE-L↑ | BLEU-4↑ |
| NaturalSigner | **44.36** | **20.07** | 43.91 | **19.70** |
| w/o Feed-Forward Denoiser | 39.25 | 16.44 | 37.31 | 15.73 |
| w/o Mixed Semantic Encoder | 42.59 | 18.76 | 42.77 | 18.31 |
| w/o Sign Language Prompt Encoder | 43.65 | 19.63 | **44.00** | 19.65 |

respectively, after introducing the sign language prompt mechanism. In addition, the multimodal dist and diversity metrics also become better, which indicates that the sign language prompt mechanism can effectively improve the quality of the generated sign language. The ablation experiments in Table 5 show that introducing the sign language prompt mechanism is also helpful for the robustness of sign language generation (back translation metric).

**Effectiveness of Sign Language Prompting Mechanisms for Zero-shot SLG.** Another interesting question is whether the sign language prompt mechanism is effective for generating unseen signer's sign language in a zero-shot SLG setting. To answer this question, we re-divide the Phoenix2014T dataset to obtain a test set with unseen signer, and then do not use the data in this test set during training. As shown in Table 6, we can see that our proposed method has certain potential for zero-shot SLG. However, the improvement in the quality of the generated sign language is still not obvious, which may be due to the small number of sign language data and signer used for training, which makes the model's generalization ability insufficient.

Table 6: Experimental results on the zero-shot SLG setting.

| Method | FID↓ | Multimodal Dist ↓ | Diversity → |
|---|---|---|---|
| Real | $0.000^{\pm.00}$ | $0.642^{\pm.00}$ | $0.753^{\pm.13}$ |
| NaturalSigner | $0.170^{\pm.18}$ | $1.108^{\pm.22}$ | $0.701^{\pm.07}$ |
| +P | $\mathbf{0.164^{\pm.14}}$ | $\mathbf{1.091^{\pm.23}}$ | $\mathbf{0.713^{\pm.07}}$ |

**Effectiveness of Mixed Semantic Encoder.** We verify the effectiveness of the mixed semantic encoder by canceling the text branch of the mixed semantic encoder. As presented in Table 5, removing the text branch from the mixed semantic encoder results in a performance drop according to the back translation metric. This suggests that incorporating supplementary textual information can enhance the quality of the prior vector, consequently improving the generated sign language.

**Effectiveness of Feed-Forward Denoiser.** To verify the effectiveness of the Feed-Forward Denoiser we proposed, we replace the Feed-Forward Denoiser with a Transformer decoder (Vaswani et al., 2017) with the same number of parameters. As shown in Table 5, after removing the Feed-Forward Denoiser, the performance of the back translation metric of the generated sign language results drops significantly. This indicates that the Feed-Forward Denoiser we designed can more effectively utilize the information of the prior vector to maintain the robustness and semantic consistency of the generated results.

## 5 CONCLUSION

This paper introduces NaturalSigner, a SLG system that employs a non-autoregressive diffusion model to synthesize natural sign pose sequences from text input. To improve the quality of the generated sign language, we propose a mixed semantic encoder that can encode both text and gloss, and a novel Feed-Forward Denoiser to efficiently and accurately generate sign language pose sequences. In addition, we propose a sign language prompt mechanism for the duration predictor and diffusion model to improve the naturalness and expressiveness of the generated sign language. Extensive experiments show that our proposed method can achieve natural and accurate SLG, and outperforms the state-of-the-art methods on generation robustness, generation quality and generation diversity. In addition, extensive ablation experiments show that each module in our proposed method is effective.

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
