# OpenReview forum: "NaturalSigner: Diffusion Models are Natural Sign Language Generator"
_ICLR.cc/2024/Conference — ICLR 2024 Conference Withdrawn Submission_

### Official Review · Reviewer_s9QL · 2023-10-30

**Soundness:** 3 good
**Presentation:** 2 fair
**Contribution:** 3 good
**Rating:** 5
**Confidence:** 4

**Summary:**

The focus of this paper is on development for improved models for signal language generation. They propose a classifier-free diffusion model that goes from gloss or text to animation and produces parameters derived from SMPL-X. Results on Phoenix-2014 and Phoenix-2014T are significantly better than competing methods according to translations metrics (Rough, BLEU-4). Qualitative results show that animations from this model are preferred over one competing model.

**Strengths:**

* The motivation behind using diffusion models is sound (e.g., overcoming the one-to-many problem)
* The use of SMPL-X seems like a nice improvement over more common key point-based approaches.
* The experiment ablations are nice and I appreciate the authors for breaking apart the prompting mechanism, semantic encoder, and deef forward denier.

**Weaknesses:**

* Related work is missing many references in the NLP, HCI, and Accessibility communities. There has been a lot of interest in SL modeling over the past 1-2 years but many of the references refer primarily to a line of work by Saunders et al. ending in 2021.
* The descriptions of the model/system formulation could benefit from more depth. Specifically, the introduction of the duration model isn't entirely clear. My hypothesis is that this is being used in the same way as duration models in TTS speech systems, but I'm not entirely sure. As an aside, given connections to TTS systems, it may be worth digging into that more in the related work section.
* I didn't quite understand the motivation behind the prompt encoder. Maybe I missed it, but it might be worth clarifying why this is used. Are the results in Table5 and 6 statistically significant? The results with prompt encoder and without are very similar and in one case (Table 5 test) better without this encoder.
* Is FID a good metric for this problem? Is there a demonstrable correlation between improved FID and improved signing quality?
* Subjective evaluation shows that this is better than Saunders, but doesn't give an overall sense of quality. Without videos it's unclear how well the approaches work (subjectively).

**Questions:**

I am on the fence about this paper. Can the authors provide video references for the animations? And answer some of the questions in the 'weakness' section?

---

> ### Author Response · Authors · 2023-11-20
> **Response to Reviewer s9QL**
>
> Thank you for acknowledging the SLG task in our paper and providing valuable feedback. We have thoroughly considered your comments and implemented important clarifications and improvements to address the raised is
>
> 1. Related work is missing many references in the NLP, HCI, and Accessibility communities. There has been a lot of interest in SL modeling over the past 1-2 years but many of the references refer primarily to a line of work by Saunders et al. ending in 2021.
>
> We will add more latest papers and discussions on the SLG field, such as SignDiff [1], to the camera ready version of the paper.
>
> 2. About duration prediction model in NaturalSigner
>
> The working principle of the duration prediction model is indeed similar to the way it works in the TTS speech system, because the purpose of their introduction is similar, that is, to predict the duration corresponding to each semantic unit, which corresponds to sign gloss in sign language and to the phoneme in TTS.
>
> 3. the motivation behind the prompt encoder
>
> **The introduction of the sign language prompt mechanism reduces the difficulty of SLG in processing one-to-many problem**. Assume that when there is no prompt, there may be M kinds of sign language poses corresponding to text. When there is prompt, there may be N kinds of sign language poses corresponding to text. Obviously M>=N (it can be known from the Pigeonhole Principle)
>
> In practical applications, it is very easy to obtain a sign language prompt video within 16 seconds. I think it is very worthwhile to pay such a small price to improve the quality of generated sign language.
>
> The reason why the use of sign language prompt mechanism in Table 5 is not particularly significant is that what we show here is the back translation metric, which uses a translation model to perform back translation. **A good translation model can correctly translate different sign language pose sequences representing the same meaning into the same text (that is, many to one), and the method we proposed can perform semantic analysis even without a sign language prompt mechanism. Accurate one to many mapping**. However, due to the existence of the translation model, they eventually become the only text during testing, so the sign language prompt mechanism is not very significant in the back translation indicator.
>
> An example is assuming that the translation model can map {A,B,C,D} to x. When the sign language prompt mechanism is not used, NaturalSigner randomly generates {A,B,C,D} based on x. When the sign language prompt mechanism is used, NaturalSigner Randomly generate {A,B} based on x. However, in the final test of the back translation index, both BLEU and ROUGE were calculated using x, so the indexes were very close.
>
> In Table 2 and Table 3, we can observe that the improvement brought about by using the sign language prompt mechanism is still very significant in terms of FID and other indicators.
>
> One reason why the setting of zero-shotSLG in Table 6 is not particularly significant is that the number of signers in the sign language data set and the sign language data are too small, making it difficult to generalize to unseen signers. A comparison is that valle [4], which has a better zero-shot effect in TTS, uses the LibriLight data set to contain 60k h of data and 7439 speakers, while the number of signers in Phoenix-2014T is 9 and contains a total of 11h of data. We believe that NaturalSigner should achieve better results when the amount of data is sufficient, which is also one of our future work.
>
> 4. is FID a good metric for this problem
>
> First of all, we believe that sign language itself is also a special form of human motion. Many previous human motion synthesis works, such as dance synthesis [2], natural motion synthesis [3], etc., all use FID as an indicator to measure the quality of generated movements. . And these works show a correlation between improved FID and improved motion quality.
>
> 5. Subjective evaluation shows that this is better than Saunders, but doesn't give an overall sense of quality. Without videos it's unclear how well the approaches work (subjectively).
>
> The video we show on the demo page shows that the sign language video generated by our method is more expressive than the sign language video generated by the baseline method. For example, the baseline method will appear static and stiff, but our method will not.
>
> We are trying to use blender to render video based on SMPLX parameters, we will provide it as soon as the video rendering is completed. (Please forgive us because testing on the new CSL-Daily dataset took a lot of time)
>
> [1] SignDiff: Learning Diffusion Models for American Sign Language Production
>
> [2] Dancing to music
>
> [3] T2M-GPT: Generating Human Motion from Textual Descriptions with Discrete Representations
>
> [4] Neural Codec Language Models are Zero-Shot Text to Speech Synthesizers

---

> ### Author Response · Authors · 2023-11-21
> **Hoping that our response could address your concern**
>
> Dear Reviewer s9QL,
>
> Thank you again for your time and effort in reviewing our work! We submitted the animations video in our latest submission of supplemental materials. We would be grateful if you could let us know if our response has addressed your concerns. As the rebuttal period draws to a close, we look forward to hearing from you and remain at your disposal for any further clarification you may require.
>
> Thanks in advance,
>
> Paper 5020 Authors

---

> ### Comment · Reviewer_s9QL · 2023-11-23
> **Post rebuttal**
>
> I have glanced through other reviews and I have looked at the rebuttal and a handful of the animation videos. I remain borderline. I appreciate how much extra effort the authors have put into this work and think it would be a reasonable paper to accept. I do not feel strongly enough to champion it against dissent from other reviews.
>
> Some aspect, such as the lack of facial animation, continue to be a concern. I also appreciate that the author ran a user study but think the ultimate results are limited given the relative scores between two models (as opposed to overall global assessment of whether or not the models were useful).

---

> > ### Author Response · Authors · 2023-11-23
> > **Response to Reviewer s9QL**
> >
> > Thank you very much for taking the time out of your busy schedule to reply to me, and I'm glad you think this is a reasonable paper worthy of acceptance. I will continue to answer your questions next.
> >
> > 1. An explanation of how important facial expressions are in sign language
> >
> > We would like to further explain how important facial expressions are in sign language. In fact, the importance of facial expressions in sign language is highly related to the usage scenario. The survey results in the paper [1] show that the addition of facial expressions brings more significant evaluation scores in the emotion category of sign language, but the improvement in non-emotion categories is limited. In addition, observing Table 6 in [1], we can also find that for the ASL-Originated setting, even in the emotion category, there can be good scores without facial expressions. These evaluation results can illustrate two points.
> >
> > - **In sign language, gestures and body motion convey the main information, and facial expressions are an important supplement.**
> > - **Facial expressions play a more important role when conveying emotional information, and play a relatively limited role in describing objective facts.**
> >
> > Therefore, it can be considered that **the sign language we generate contains the main information that needs to be expressed in sign language**, and the method we propose itself can also be seamlessly extended to include facial expressions once the facial expression parameters in SMPLX are available. Another thing worth noting is that the theme of Phoenix-2014T data set is weather forecast, and the theme of CSL-Daily is daily life (such as travel, shopping, medical care). **These datasets mainly describe objective facts and contain less emotional information. So for such datasets and scenarios, facial expressions may not be that important** from the perspective of conveying information and communication, although of course it is always better to include facial expressions.
> >
> > 2. About the evaluation results
> >
> > Due to the lack of open source code and the difficulty of user study, it is difficult for us to evaluate all methods. However, we believe that the evaluation results of the objective metrics we provide can provide an absolute score for a fair comparison of all methods. We will also publish our model and evaluation code to facilitate future researchers to calculate corresponding evaluation scores.
> >
> > [1]  [Evaluating Facial Expressions in American Sign Language Animations for Accessible Online Information | SpringerLink](https://link.springer.com/chapter/10.1007/978-3-642-39188-0_55)
> >
> >
> >
> > ------
> >
> > We strive to provide quick and comprehensive explanations for your inquiries, and we believe that our responses will reassure you and deepen your understanding of the motivation behind this work.As the end of the author-reviewer discussion period is approaching, if you have any additional questions or require further clarification, please feel free to ask. We would greatly appreciate a prompt response.
> >
> >
> >
> > Best regards, Authors

---

### Official Review · Reviewer_9cnq · 2023-10-30

**Soundness:** 3 good
**Presentation:** 3 good
**Contribution:** 3 good
**Rating:** 8
**Confidence:** 5

**Summary:**

In this paper, the authors propose a diffusion model based SLG approach. The presented model uses SMPL-X body parameters as sign representations, and is able to generate realistic signer poses when being prompted with sign gloss and spoken language text. The authors conduct extensive experiments on the Phoenix datasets and report significantly better back-translation and FID scores compared to the state-of-the-art.

**Strengths:**

- Although diffusion models have been used for SLG before (https://arxiv.org/pdf/2308.16082v1.pdf) for photo realistic avatar generation, this paper is the first application of diffusion models to generate sequence of signer poses given sign glosses and spoken language text, to the best of my knowledge.
- The authors are sharing their source code.
- The proposed approach achieves significantly better back-translation and FID scores on the Phoenix2014 datasets compared to the state-of-the-art.
- The authors conduct a user study with 10 participants to qualitatively assess their approach. I should stress the value of this, as the CV community does not commonly conduct user studies.
- Ablation study was informative.

**Weaknesses:**

- Not including facial expressions was disappointing. As the authors would appreciate, to fully convey and understand meaning of sign language utterances one must consider facial expressions, mouthings, gaze and mouth gestures.
- Given the authors use a parametric body model as their sign representation, it was quite surprising to see that they've chosen skeletons as their visualization instead of a canonical body being driven by the generated pose configurations.
- Although popular, Phoenix datasets are quite limited in terms of domain and signing variance. I'd have greatly strengthened the paper if the authors conducted studies in other datasets, like OpenASL or CSL, to set baselines for future research.

**Questions:**

Q: User Study: Were the participants deaf and proficient in DGS? I am asking since the authors mention that the participants evaluate the approaches based on the generated sequence being "easier to understand and closer to the ground truth”. Please clarify this in the manuscript.

Suggestions and Minor Fixes:
- Please use the term "Deaf and Hard of Hearing" instead of "Hearing-impaired"
- Use “ “ instead `` ‘’ in the latex to fix the quotation mark issues.
- Page 2 naturalSigner -> NaturalSigner

---

> ### Author Response · Authors · 2023-11-20
> **Response to Reviewer 9cnq**
>
> Thank you for acknowledging the SLG task in our paper and providing valuable feedback. We have thoroughly considered your comments and implemented important clarifications and improvements to address the raised issues:
>
> 1. **Why does the generated sign language pose not include facial expressions?**
>
> Thanks to the review for pointing this out. Facial expressions are indeed very important in sign language expression. T**he lack of facial expressions in sign language generation results is indeed a common problem in SLG tasks. In fact, the results of almost all previous work on sign language generation do not contain facial keypoints.** [1] [6] [7] [8] [9]
>
> The generation of facial keypoints itself is a very challenging task, as a separate field called talking face generation. **Generating accurate facial keypoints requires a large amount of data.** For example, [10] used **the LRS3-TED dataset** to train a model that generates facial key points, which contains more **than 400h of video data**. The **Phoenix-2014T data set only has 11h of data.**
>
> **Therefore, previous work mainly focused on the synthesis of hand poses, which is easier and more important, and temporarily ignored the face.**
>
> **Another potential limitation is that the frankmocap toolbox we used cannot extract the facial parameters of SMPLX.**
>
> In the future we plan to try to use tools such as [vchoutas/smplify-x: Expressive Body Capture: 3D Hands, Face, and Body from a Single Image (github.com)](https://github.com/vchoutas/smplify-x) Box extracts SMPLX facial parameters and attempts to transfer facial joint point synthesis capabilities from the talking face generation field to the SLG field (due to the commonality between facial expressions)
>
> 2. Given the authors use a parametric body model as their sign representation, it was quite surprising to see that they've chosen skeletons as their visualization instead of a canonical body being driven by the generated pose configurations.
>
> Since the previous models [1] [7] [8] [9] and baseline [1] models all use 3D key points as the representation of sign language, in order to facilitate comparison with them, we rewrite the SMPLX model and map it to 3D key points. Click to compare with them.
>
> Furthermore, we are trying to use blend to render video based on SMPLX parameters, we will provide it as soon as the video rendering is completed. (Please forgive us because testing on the new CSL-Daily dataset took a lot of time)
>
> 3. Although popular, Phoenix datasets are quite limited in terms of domain and signing variance. I'd have greatly strengthened the paper if the authors conducted studies in other datasets, like OpenASL or CSL, to set baselines for future research.
>
> We supplement the test results on the CSL-Daily dataset in the Response to all reviewers section.
>
> [1] Progressive transformers for end-to-end ¨ sign language production
>
> [2] Diffwave: A versatile diffusion model for audio synthesis
>
> [3] Video diffusion models
>
> [4] DiffSinger: Singing Voice Synthesis via Shallow Diffusion Mechanism
>
> [5] Person Image Synthesis via Denoising Diffusion Model
>
> [6] SignDiff: Learning Diffusion Models for American Sign Language Production
>
> [7] Adversarial training for multi-channel ¨ sign language production
>
> [8]  Continuous 3D multi-channel sign language production via progressive transformers and mixture density networks.
>
> [9] Mixed SIGNals: Sign Language Production via a Mixture of Motion Primitives
>
> [10] GeneFace: Generalized and High-Fidelity Audio-Driven 3D Talking Face Synthesis

---

### Official Review · Reviewer_UZBd · 2023-10-31

**Soundness:** 2 fair
**Presentation:** 2 fair
**Contribution:** 2 fair
**Rating:** 3
**Confidence:** 5

**Summary:**

The paper proposes to use a diffusion model to generate sign language keypoints. More specifically, the authors first design a sign language prompting mechanism considering the signer identity inforation. Then a mixed semantics encoder considering both text, gloss, and prompt, and a duration predictor are proposed as a prior model. Finally, a diffusion process is achieved as usual. The overall method achieves new SOTA performance on a series of metrics.

**Strengths:**

1. Use SMPLX parameters to represent motion is reasonable.
2. Sign languge prompting considers signer identity information.
3. SOTA performance on Phoenix-2014T on multiple metrics.
4. A user study is conducted.

**Weaknesses:**

1. The completeness of the paper is low.

a) I don't think generated keypoints are understandable to the deaf. Facial expression and mouth movement are important to sign language understanding but they are not included as shown in the demo. The generated keypoints may be better than baselines, but they are still far from being understood by the deaf people.

b) A classic prior work, FS-Net [1], has already achieved **video** generation using the keypoints to drive a signer image. But the proposed method doesn't support video generation.

c) Similarly, a recent open-sourced work, SignDiff [2], also applies the diffusion model on sign language generation, while it also provides video results.

2. The novelty is limited. The paper is more like an application of diffusion model on sign language generation, while the applicability has already been verified in SignDiff [2]. Although some adaptations are proposed, some of them, e.g., duration predictor, already appear in existing sign language papers [3,4].

3. It is good to involve signer identity information in the sign langauge prompting, but there are not corresponding experiments to verify its effectiveness. I don't think current generated results can reflect signer identity information.

4. Need more details for the user study. Since the task is for the deaf people, how are they fluent with sign language? Are they Germany sign language users? How many videos are given to the users? Are the text and gloss annotations given to them?

5. An important benchmark, CSL-Daily, is missing. Phoenix-2014 and Phoenix-2014T are quite similar, and thus the conclusions on these two benchmarks are always consistent. Other benchmarks from a different language, e.g., CSL-Daily, is necessary.

[1] Signing at Scale: Learning to Co-Articulate Signs for Large-Scale Photo-Realistic Sign Language Production, CVPR 2022

[2] SignDiff: Learning Diffusion Models for American Sign Language Production, arXiv 2023

[3] SimulSLT: End-to-End Simultaneous Sign Language Translation, MM 2021

[4] Towards Fast and High-Quality Sign Language Production, MM 2022

**Questions:**

See weakness.

---

> ### Author Response · Authors · 2023-11-20
> **Response to Reviewer UZBd**
>
> Thank you for acknowledging the SLG task in our paper and providing valuable feedback. We have thoroughly considered your comments and implemented important clarifications and improvements to address the raised issues:
>
> 1. **Why does the generated sign language pose not include facial expressions?**
>
> Thanks to the review for pointing this out. Facial expressions are indeed very important in sign language expression. T**he lack of facial expressions in sign language generation results is indeed a common problem in SLG tasks. In fact, the results of almost all previous work on sign language generation do not contain facial keypoints.** [1] [6] [7] [8] [9]
>
> The generation of facial keypoints itself is a very challenging task, as a separate field called talking face generation. **Generating accurate facial keypoints requires a large amount of data.** For example, [10] used **the LRS3-TED dataset** to train a model that generates facial key points, which contains more **than 400h of video data**. The **Phoenix-2014T data set only has 11h of data.**
>
> **Therefore, previous work mainly focused on the synthesis of hand poses, which is easier and more important, and temporarily ignored the face.**
>
> **Another potential limitation is that the frankmocap toolbox we used cannot extract the facial parameters of SMPLX.**
>
> In the future we plan to try to use tools such as [vchoutas/smplify-x: Expressive Body Capture: 3D Hands, Face, and Body from a Single Image (github.com)](https://github.com/vchoutas/smplify-x) Box extracts SMPLX facial parameters and attempts to transfer facial joint point synthesis capabilities from the talking face generation field to the SLG field (due to the commonality between facial expressions)
>
> 2. Why are no sign language video results generated?
>
> **Our work mainly focuses on generating sign language pose sequences from sign glosses and spoken language text**, which are crucial for sign language generation. **Generating sign language videos based on sign pose sequences is not the focus of this article**, because this part actually has nothing to do with the semantics of sign language and belongs to the field of video to video synthesis.
>
> This part of the generation work can be completed by many existing tasks, such as SignDiff[6], ControlNET[10], and Vid2Vid[11]. In fact, such tasks as ControlNET and Vid2Vid do not even rely on sign language data. SignDiff It also benefits from training on a large amount of ordinary human posture data.
>
> 3. **Comparison with SignDiff**
>
> We have the following points to clarify about the signdiff paper:
>
> - **The submission time of this paper (2023.08) is very close to the submission time of our paper (2023.09). Within 3 months, we did not notice this work when we were writing.**
> - **SignDiff has only been submitted on arxiv and has not passed peer review, which means that the completeness and reference value of this paper may be low.**
> - **SignDiff does not conflict with our work. It can be used as a downstream tool chain for our work to generate sign language videos. Because signdiff mainly tries to use the diffusion model in sign pose to sign vidoe task.**
>
> 4. **About the effectiveness of the proposed sign language prompt mechanism**
>
> Our experiments in 4.3 ABLATION STUDY proved the effectiveness of the sign language prompt mechanism.
>
> **As shown in Table 2 and Table 3, on the Phoenix2014T and Phoenix2014 datasets, the FID metrics decreased by more than 33% and 17%.**
>
> In fact, **the introduction of the sign language prompt mechanism reduces the difficulty of SLG in processing one-to-many problem**. Assume that when there is no prompt, there may be M kinds of sign language poses corresponding to text. When there is prompt, there may be N kinds of sign language poses corresponding to text. Obviously M>=N  (it can be known from the Pigeonhole Principle)
>
> In practical applications, it is very easy to obtain a sign language prompt video within 16 seconds. I think it is very worthwhile to pay such a small price to improve the quality of generated sign language.
>
> [1] Progressive transformers for end-to-end ¨ sign language production
>
> [2] Diffwave: A versatile diffusion model for audio synthesis
>
> [3] Video diffusion models
>
> [4] DiffSinger: Singing Voice Synthesis via Shallow Diffusion Mechanism
>
> [5] Person Image Synthesis via Denoising Diffusion Model
>
> [6] SignDiff: Learning Diffusion Models for American Sign Language Production
>
> [7] Adversarial training for multi-channel ¨ sign language production
>
> [8]  Continuous 3D multi-channel sign language production via progressive transformers and mixture density networks.
>
> [9] Mixed SIGNals: Sign Language Production via a Mixture of Motion Primitives
>
> [10] Adding Conditional Control to Text-to-Image Diffusion Models
>
> [11] Video-to-Video Synthesis
>
> [12] Neural Codec Language Models are Zero-Shot Text to Speech Synthesizers

---

> ### Author Response · Authors · 2023-11-21
> **Hoping that our response could address your concern**
>
> Dear Reviewer UZBd,
>
> Thank you again for your time and effort in reviewing our work! We further added evaluation results on the CSL-Daily dataset and submitted animations video in the latest supplementary material submission. We would be grateful if you could let us know if our response has addressed your concerns. As the rebuttal period draws to a close, we look forward to hearing from you and remain at your disposal for any further clarification you may require.
>
> Thanks in advance,
>
> Paper 5020 Authors

---

> > ### Comment · Reviewer_UZBd · 2023-11-23
> >
> > Thank the authors for their rebuttal. The comparison with SignDiff is persuasive. However, I have more concerns after reading the authors' rebuttal.
> >
> > 1. The back-translation performance on CSL-Daily is questionable. Why is it so good? It can even achieve a BLEU-4 score of more than 40! To the best of my knowledge, the SOTA work (TwoStream Network) can only achieve a BLEU-4 score of 25.79 on CSL-Daily in a fully-supervised setting using a heavy two-stream model. Why a "back-translation" model can even double the BLEU-4 score?
> >
> > 2. It is pretty good to include animation results. They are much better than keypoints. However, it is really troublesome to compare the results with GT and baseline because the authors put them into separate folder. For each sample, I suggest the authors to put them into one video side by side to ease comparison.
> >
> > 3. I can only check several samples since it is quite difficult to compare them. The major issue is that I cannot conclude the proposed method perform well. This is mainly because there are not SMPLX annotations for current sign language datasets. The term, GT, is misleading. I think the authors should use raw videos instead of a virtual human to represent GT.
> >
> > Given these new concerns, I cannot increase my rating. Hope that addressing these concerns can further improve the paper.

---

> ### Author Response · Authors · 2023-11-23
> **Response to Reviewer UZBd**
>
> Thank you very much for taking the time to reply to me in your busy schedule. I am glad that we have answered your concerns related to SignDiff. Next I will continue to answer your questions.
>
> 1. Details about back translation evaluation metrics
>
> Maybe you misunderstood how the back translation indicator is calculated. We will provide further clarification on this.  As we describe in the back translation section of Appendix C EVALUATION METRICS summary, similar to previous work [1-4] we use all data to train a back translation sign language translation model.  The ROUGE and BLEU4 of the back translation model on the ground truth are 99, so the ROUGE and BLEU4 scores of the test will be higher than the translation scores of the "TwoStream Network" [5]. This model is only used during evaluation so there is no risk of data leakage. We will further emphasize this point in the camera ready version of the paper to avoid misunderstandings.
>
> 2. About demo display method
>
> Thanks for your suggestion, we realize that the current presentation method does cause confusion for comparisons. We merge the original video, ground truth, NaturalSigner, Progressive transformers animations video into the same video and resubmit it to the supplementary material. Hopefully our new video submission will make it easier for you to compare. We can find that the videos generated by our NaturalSigner (third column) are much better than the baseline model.
>
> ----
> We strive to provide quick and comprehensive explanations for your inquiries, and we believe that our responses, coupled with the newly supplemented combined demo video, will reassure you and deepen your understanding of the motivation behind this work.
> As the end of the author-reviewer discussion period is approaching, if you have any additional questions or require further clarification, please feel free to ask. We would greatly appreciate a prompt response.
>
>
>
> Best regards, Authors

---

### Official Review · Reviewer_BJCa · 2023-11-05

**Soundness:** 2 fair
**Presentation:** 2 fair
**Contribution:** 2 fair
**Rating:** 3
**Confidence:** 5

**Summary:**

This paper studies the topic of sign language generation. It proposes a NaturalSigner framework to leverage the strong modeling capability of diffusion model. The experiments on two datasets demonstrate the effectiveness of the proposed method.

**Strengths:**

This paper studies the topic of sign language generation. It proposes a NaturalSigner framework to leverage the strong modeling capability of the diffusion model.

The experiments on two datasets demonstrate the effectiveness of the proposed method.

**Weaknesses:**

One of the main concerns is that the presented qualitative results are not satisfying. For the keyframe in Figure 4/video on the demo webpage, I do not see a clear improvement over the previous method.

The title is confusing. What does the word natural mean?

The novelty is somewhat limited. The authors should clearly state why diffusion better works for SLG.

I do not think section 3.2 should be called in-context learning. It is just an embedding technique.

What is the specific setting of zero-shot SLG?

**Questions:**

The questions are listed above.

---

> ### Author Response · Authors · 2023-11-20
> **Response to Reviewer BJCa**
>
> Thank you for acknowledging the SLG task in our paper and providing valuable feedback. We have thoroughly considered your comments and implemented important clarifications and improvements to address the raised issues:
>
> 1. One of the main concerns is that the presented qualitative results are not satisfying. For the keyframe in Figure 4/video on the demo webpage, I do not see a clear improvement over the previous method.
>
> In the video on the demo page, we can clearly observe that the video generated by the baseline model on the right has stiff and static motions. This is because regression-based SLG has difficulty processing one-to-many mapping and generates over-smooth sign language gestures. sequence. However, the results generated by our model (middle part) do not have this problem, and the actions are closer to the ground truth.
>
> 2. The title is confusing. What does the word natural mean?
>
> We named it NaturalSigner because the FID, Multimodal Dist and Diversity metrics indicate that the sign language we generate is more expressive, and at the same time closer in features to the real natural sign language (FID), and clearer in semantic representation (Multimodal Dist and back translation)
>
> 3. The novelty is somewhat limited. The authors should clearly state why diffusion better works for SLG.
>
> Diffusion models have achieved success in various domains such as speech synthesis [2], video synthesis [3], singing voice synthesis [4], and person image synthesis [5]. These models have demonstrated desirable properties, including **ease of training and the ability to effectively capture many-to-many matching distributions.** However, the application of diffusion models in the Sign Language Generation (SLG) task has not been explored yet. Additionally, as described in the first paragraph of our introduction section, **SLG tasks face challenges related to one-to-many mapping**. Therefore, we propose that diffusion models are better candidates for SLG and introduce a novel approach called NaturalSigner.
>
> As reviewer 9cnq pointed out, **this paper represents the first application of diffusion models to generate a sequence of signer poses based on sign glosses and spoken language tex**t. It is important to clarify that **the main focus of the SignDiff work is to modify ControlNet in order to generate realistic sign language videos based on a sequence of sign poses**. Therefore, it is not conflicting with this paper; in fact, both works can be seamlessly connected to form a complete pipeline.
>
> We can highlight the novelty of this approach from the following points:
>
> - **We develop the first diffusion-based approach for sign language pose generation task which, which can generate accurate and expressive sign language pose sequences.**
>
> - **To improve the semantic consistency and expressiveness of the generated sign language, we propose a novel mixed semantic encoder and sign language prompt mechanism to generate more accurate prior vectors.**
>
> - **A carefully designed Feed-Forward Denoiser is used to maintain the semantic consistency between the generated results and the prior vectors in the time dimension while reducing the computational complexity.**
>
> - **Extensive experiments on three datasets show the effectiveness of our proposed method. The inclusion of additional evaluation metrics and a broad range of new baseline results can provide valuable guidance for future research in this field.**
>
> 4. I do not think section 3.2 should be called in-context learning. It is just an embedding technique.
>
> Maybe my understanding is wrong. I think in-context learning means that the model can learn by adding examples to the input without updating the model parameters. The sign language prompt mechanism we proposed can generate similar sign languages according to the sign language prompt without updating the model parameters, which should meet this definition.
>
> 5. What is the specific setting of zero-shot SLG?
>
> For zero-shot SLG, we re-divided Phoenix-2014T according to sign and retained an unseen signer as a test set to test the model's ability to zero-shot sign language generation on unseen signers.

---

> ### Author Response · Authors · 2023-11-21
> **Hoping that our response could address your concern**
>
> Dear Reviewer BJCa,
>
> Thank you again for your time and effort in reviewing our work! We further add the evaluation results on the CSL-Daily dataset, submit an animations video in the latest supplementary material submission and further explain the novelty of our method. We would be grateful if you could let us know if our response has addressed your concerns. As the rebuttal period draws to a close, we look forward to hearing from you and remain at your disposal for any further clarification you may require.
>
> Thanks in advance,
>
> Paper 5020 Authors

---

### Author Response · Authors · 2023-11-20
**Response to all reviewers**

We express our gratitude to the reviewers for recognizing various aspects of our work:

- **Use SMPLX parameters to represent motion is reasonable**（ UZBd，s9QL）
- **Sign languge prompting considers signer identity information.**（UZBd）
- **SOTA performance on Phoenix-2014T and Phoenix-2014 dataset on multiple metrics.** （BJCa，UZBd，9cnq）
-  **It is reasonable to use diffusion models in SLG and the first application of diffusion models to generate sequence of signer poses given sign glosses and spoken language text**（BJCa，9cnq，s9QL）
- **A user study is conducted.**（UZBd，9cnq）
- **The authors are sharing their source code.**（9cnq）
- **Ablation study was informative.**（9cnq，s9QL）

We are also thankful for the valuable suggestions provided by the reviewers, and in the following, we will highlight and clarify a few essential points:

1. **Supplementary experimental evaluation results on the CSL-Daily dataset**

Thank you for the suggestions provided by reviewers UZBd and 9cnq. Testing the SLG task on sign language benchmarks in different languages is indeed meaningful. We have conducted tests on the baseline model as well as our proposed model using the CSL-Daily dataset, a large-scale Chinese sign language dataset. The test results are presented below:

Back translation results for the Gloss to Pose task on DEV part.

|                       | ROUGE↑    |  BLEU1↑   | BLEU2↑    | BLEU3↑    | BLEU4↑    |
| --------------------- | --------- | :-------: | --------- | --------- | --------- |
| PT                    | 52.02     |   55.82   | 46.98     | 42.07     | 39.72     |
| NaturalSigner（ours） | **64.48** | **62.64** | **55.70** | **52.04** | **49.63** |

Back translation results for the Gloss to Pose task on TEST part.

|                       | ROUGE↑    |  BLEU1↑   | BLEU2↑    | BLEU3↑    | BLEU4↑    |
| --------------------- | --------- | :-------: | --------- | --------- | --------- |
| PT                    | 50.42     |   54.89   | 44.57     | 38.83     | 36.67     |
| NaturalSigner（ours） | **62.75** | **62.84** | **55.28** | **50.82** | **47.67** |

Back translation results for the Text to Pose task on DEV part.

|                       | ROUGE↑    |  BLEU1↑   | BLEU2↑    | BLEU3↑    | BLEU4↑    |
| --------------------- | --------- | :-------: | --------- | --------- | --------- |
| PT                    | 52.94     |   56.38   | 47.06     | 41.82     | 38.57     |
| NaturalSigner（ours） | **64.75** | **64.33** | **56.88** | **52.39** | **48.35** |

Back translation results for the Text to Pose task on TEST part.

|                       | ROUGE↑    |  BLEU1↑   | BLEU2↑    | BLEU3↑    | BLEU4↑    |
| --------------------- | --------- | :-------: | --------- | --------- | --------- |
| PT                    | 49.90     |   53.31   | 43.20     | 37.64     | 35.34     |
| NaturalSigner（ours） | **61.65** | 59.81**** | **52.24** | **47.97** | **45.62** |

Experimental results the Gloss to Pose task on the CSLDaily dataset on DEV part.(± indicates the 95% confidence
interval.The range of the interval is multiplied by 100 for ease of data display.)

|               | FID↓          | Multimodal Dist↓ | Diversity→    |
| ------------- | ------------- | ---------------- | ------------- |
| Real          | 0.000±.00     | 0.773±.00        | 0.315±.06     |
| PT            | 1.152±.00     | 1.221±.00        | 0.004±.00     |
| NaturalSigner | 0.078±.06     | 0.915±.08        | 0.307±.03     |
| +P            | **0.063±.05** | 0.873±.02****    | **0.309±.03** |

Experimental results the Gloss to Pose task on the CSLDaily dataset on TEST part.

|               | FID↓          | Multimodal Dist↓ | Diversity→    |
| ------------- | ------------- | ---------------- | ------------- |
| Real          | 0.000±.00     | 0.774±.00        | 0.287±.06     |
| PT            | 1.879±.00     | 1.408±.00        | 0.076±.12     |
| NaturalSigner | 0.082±.05     | 0.915±.08        | **0.281±.02** |
| +P            | **0.069±.03** | **0.861±.11**    | 0.280±.01     |

It can be seen that on the large Chinese sign language data set of CSL-Daily, the method we proposed is significantly better than the baseline model PT in terms of back translation, FID, Multimodal Dist and Diversity. This shows that our proposed method is more accurate and expressive than the sign language generated by the baseline model. We will cite the original paper of the CSL-Daily dataset and include the additional experimental results in the camera-ready version of the paper to guide future developments in this field, if the paper is ultimately accepted.

---

> ### Author Response · Authors · 2023-11-20
> **Response to all reviewers**
>
> 2 **Details about user study**
>
> Each participating user is proficient in DGS . We provide each user with ground truth video references, but we do not provide reference sign language glosses and text. Instead, we randomly shuffle the results generated by the baseline model and our proposed model, providing users with 20 examples per dataset to choose from.
>
>
> 3 **About the novelty of NaturalSigner and the rationality of using the diffusion model in SLG**
>
> Diffusion models have achieved success in various domains such as speech synthesis [2], video synthesis [3], singing voice synthesis [4], and person image synthesis [5]. These models have demonstrated desirable properties, including **ease of training and the ability to effectively capture many-to-many matching distributions.** However, the application of diffusion models in the Sign Language Generation (SLG) task has not been explored yet. Additionally, as described in the first paragraph of our introduction section, **SLG tasks face challenges related to one-to-many mapping**. Therefore, we propose that diffusion models are better candidates for SLG and introduce a novel approach called NaturalSigner.
>
> As reviewer 9cnq pointed out, **this paper represents the first application of diffusion models to generate a sequence of signer poses based on sign glosses and spoken language tex**t. It is important to clarify that **the main focus of the SignDiff work is to modify ControlNet in order to generate realistic sign language videos based on a sequence of sign poses**. Therefore, it is not conflicting with this paper; in fact, both works can be seamlessly connected to form a complete pipeline.
>
> We can highlight the novelty of this approach from the following points:
>
> - **We develop the first diffusion-based approach for sign language pose generation task which, which can generate accurate and expressive sign language pose sequences.**
>
> - **To improve the semantic consistency and expressiveness of the generated sign language, we propose a novel mixed semantic encoder and sign language prompt mechanism to generate more accurate prior vectors.**
> - **A carefully designed Feed-Forward Denoiser is used to maintain the semantic consistency between the generated results and the prior vectors in the time dimension while reducing the computational complexity.**
> - **Extensive experiments on three datasets show the effectiveness of our proposed method. The inclusion of additional evaluation metrics and a broad range of new baseline results can provide valuable guidance for future research in this field.**

---

> > ### Author Response · Authors · 2023-11-20
> > **Response to all reviewers**
> >
> > 4. **Why does the generated sign language pose not include facial expressions?**
> >
> > Thanks to the review for pointing this out. Facial expressions are indeed very important in sign language expression. T**he lack of facial expressions in sign language generation results is indeed a common problem in SLG tasks. In fact, the results of almost all previous work on sign language generation do not contain facial keypoints.** [1] [6] [7] [8] [9]
> >
> > The generation of facial keypoints itself is a very challenging task, as a separate field called talking face generation. **Generating accurate facial keypoints requires a large amount of data.** For example, [10] used **the LRS3-TED dataset** to train a model that generates facial key points, which contains more **than 400h of video data**. The **Phoenix-2014T data set only has 11h of data.**
> >
> > **Therefore, previous work mainly focused on the synthesis of hand poses, which is easier and more important, and temporarily ignored the face.**
> >
> > **Another potential limitation is that the frankmocap toolbox we used cannot extract the facial parameters of SMPLX.**
> >
> > In the future we plan to try to use tools such as [vchoutas/smplify-x: Expressive Body Capture: 3D Hands, Face, and Body from a Single Image (github.com)](https://github.com/vchoutas/smplify-x) Box extracts SMPLX facial parameters and attempts to transfer facial joint point synthesis capabilities from the talking face generation field to the SLG field (due to the commonality between facial expressions)
> >
> > We express our sincere gratitude to all the reviewers for their diligence and invaluable suggestions. We look forward to any further insights or inquiries that may arise from our work.
> >
> > [1] Progressive transformers for end-to-end ¨ sign language production
> >
> > [2] Diffwave: A versatile diffusion model for audio synthesis
> >
> > [3] Video diffusion models
> >
> > [4] DiffSinger: Singing Voice Synthesis via Shallow Diffusion Mechanism
> >
> > [5] Person Image Synthesis via Denoising Diffusion Model
> >
> > [6] SignDiff: Learning Diffusion Models for American Sign Language Production
> >
> > [7] Adversarial training for multi-channel ¨ sign language production
> >
> > [8]  Continuous 3D multi-channel sign language production via progressive transformers and mixture density networks.
> >
> > [9] Mixed SIGNals: Sign Language Production via a Mixture of Motion Primitives
> >
> > [10] GeneFace: Generalized and High-Fidelity Audio-Driven 3D Talking Face Synthesis

---

### Author Response · Authors · 2023-11-21
**animations video now available!**

We submit animation videos rendered in blender software using SMPLX model parameters in the supplementary materials. Due to the size limitation of the supplementary material, we randomly sampled demo videos from the results of the test dataset, with the random number set to 42, and ultimately we selected 160 videos for each model. The results of ground truth, NaturalSigner (ours), and Progressive transformers are saved in the gt, NaturalSigner, and PT subfolders respectively. The annotation.txt file saves the text information corresponding to the selected video. The first column is the video id, the second column is glosssign, and the third column is spoken language text. Through the comparison of animation video results, we can intuitively feel that the generated results of our model are more natural and expressive.

---

> ### Author Response · Authors · 2023-11-23
> **The merged animations video is now available!**
>
> According to Reviewer UZBd's suggestion, we cropped the generated videos to the size of the original videos and merged them into the same video to facilitate comparison. The merged video is available in the latest supplementary material. From left to right in the video are the original video, ground truth, NaturalSigner, and Progressive transformers animations video. We can find that the videos generated by our NaturalSigner (third column) are much better than the baseline model.